# Patterns and predictors of adherence to follow-up health guidance invitations in a general health check-up program in Japan: A cohort study with an employer-sponsored insurer database

**Yuichiro Mori[1], Kunihiro Matsushita[2], Kosuke Inoue[3], Shingo Fukuma[1]***

**1** Department of Human Health Sciences, Graduate School of Medicine, Kyoto University, Kyoto, Japan,
**2** Department of Epidemiology, Johns Hopkins Bloomberg School of Public Health, Baltimore, MD, United States of America, **3** Department of Social Epidemiology, Graduate School of Medicine, Kyoto University, Kyoto, Japan

* fukuma.shingo.3m@kyoto-u.ac.jp

**Data Availability Statement:** Statistical code: https://doi.org/10.5281/zenodo.7711073(CC-BY-SA). Data set: The datasets generated and/or

## Abstract

### Background

Japan has conducted a nationwide annual health check-up program since 2008, focusing on metabolic syndrome and subsequent health guidance in individuals at high risk for cardiovascular disease. However, the adherence rate to health guidance invitations was assumed to be low in previous reports. Therefore, this study aimed to characterize adherence patterns in the program and identify major predictors of adherence to health guidance invitations.

### Methods

We studied 186,316 adults (aged 40–74 years) who were included in a nationwide employer-sponsored insurer's database in Japan at the beginning of the fiscal year 2017. We first described adherence to health check-ups, the proportion of individuals with high cardiovascular risk, and adherence to health guidance invitations. Predictors of adherence to the invitation were then identified among eligible high-risk individuals.

### Results

In 2017, 71.7% of the study population (n = 133,573) underwent health check-ups, among whom 23.2% (n = 30,979) were invited for health guidance because of their high cardiovascular risk. Among those individuals, 35.2% (n = 10,614) received health guidance. Predictors of improved adherence to health guidance invitation were older age, more concerning blood pressure or laboratory data results, and self-reported motivation for a lifestyle change.

analyzed during the current study are not publicly available because they are third-party data with a confidentiality contract. The database was provided by the Health Insurance Association for Architecture and Civil Engineering companies, Japan, and contains insured members' demographic characteristics, medical history, and laboratory testing data from health check-ups, self-reported lifestyle, lifestyle consultation records, and clinic visit records. Due to the contract with the data provider, we cannot publicize any subset of the dataset. Upon receiving requests to share a dataset or a subset of it for replicating statistical analysis (contact via the data provider's data inquiry desk: data.dokenpo@gmail.com), the data provider will make a decision regarding each request.

**Funding:** This work was supported by the Japan Society for the Promotion of Science (JSPS) KAKENHI Grant Number 19H03870. The funder played no role in this study. SF received research grant support from SOMPO Health Support, outside the submitted work.

**Competing interests:** The authors have declared that no competing interests exist.

## Conclusion

Though 70% of eligible adults attended Japan's annual cardiovascular risk check-ups, only 35% of individuals with high cardiovascular risk adhered to health guidance invitations. Future policy reforms to improve adherence to this program should target younger individuals and those with mild stages of hypertension, diabetes, or dyslipidemia.

## Introduction

Despite the huge efforts to prevent cardiovascular disease (CVD) in the past several decades, CVD remains a global public health concern, accounting for 32% of deaths worldwide in 2019 [1]. Therefore, early identification of individuals with major risk factors for CVD is a cornerstone for the effective and efficient prevention of CVD. Accordingly, the Japanese government implemented an annual health check-up program for middle-aged or older adults (40–74 years) in 2008 [2]. The health check-up program comprises two stages: (1) a physical examination and laboratory testing and (2) follow-up health guidance (consisting of either or both a lifestyle consultation and a recommended visit to a clinic) if identified as having a high CVD risk (detailed below).

Recently, a systematic review raised concerns about the effectiveness of health check-ups in the general population [3]. Similarly, several investigations on Japan's general health check-up program found limited effectiveness in CVD outcomes [4–6]. A study pointed out that the limited effectiveness is partly due to the low adherence to health guidance invitations. Individuals who received health guidance showed significant body weight loss, while 15.9% of eligible participants in the study received lifestyle consultations [4]. However, barriers to adherence to follow-up health guidance invitations in the program are unknown because previous investigations focused on adherence to attending health check-ups [7–11].

Therefore, it is vital that detailed adherence patterns of this program are described, and any barriers to adherence to health guidance invitations are identified. By using health claims data provided by one of the largest employer-sponsored insurer in Japan, we aimed to characterize the adherence patterns of the annual CVD risk evaluation program for the general population in Japan and identify the major predictors of adherence to health guidance invitations.

## Materials and methods

The present study was reported in accordance with the Strengthening the Reporting of Observational Studies in Epidemiology (STROBE) guidelines [12] (**see S1 Table**).

### National health check-up program in Japan

Japan's national health check-up program is a government-led nationwide screening program (*Tokutei Kenshin* in Japanese) that examines CVD risk factors in middle-aged and older adults (40–74 years), with a particular focus on metabolic syndromes [2]. In Japan, every citizen is covered by one of the government-certified insurers [13] who are obligated to urge insured members to attend health check-ups conducted by hospitals or clinics, as reported previously [4,7,14,15]. In addition, if the members are employed, the employers are also obliged to provide them with the opportunity to attend health check-ups.

At the beginning of a fiscal year (from April to March in Japan), insurers provide eligibility to receive health check-ups and determine the date of attendance for individuals who have a

birthday to be 40–75 years old in the fiscal year. Individuals at 74 years old at the beginning of the fiscal year should receive health check-ups before their 75th birthday. All individuals who underwent health check-ups received their results via mail. A high CVD risk was defined as obesity (body mass index [BMI] $\geq$ 25 kg/m$^2$ or waist circumference $\geq$ 85 cm in men or $\geq$ 90 cm in women) plus at least one of the following conditions: systolic blood pressure (SBP) $\geq$ 130 mmHg or diastolic blood pressure (DBP) $\geq$ 85 mmHg; fasting blood glucose (FBG) $\geq$ 100 mg/dL or hemoglobin A1c (HbA1c) $\geq$ 5.6%; triglyceride (TG) $\geq$ 150 mg/dL or high-density lipoprotein cholesterol (HDL) $\leq$ 40 mg/dL [16]. If participants were already taking prescribed medication for hypertension (HTN), diabetes mellitus (DM), or dyslipidemia (based on self-reporting), they were instructed to continue routine clinic visits. Otherwise, individuals meeting the definition of high CVD risk receive subsequent invitations by insurers to receive lifestyle consultations or visit clinics.

### Additional interventions for selected high-risk individuals to urge them to receive health guidance

Insurers were allowed to design and conduct additional interventions for selected high-risk individuals to urge them to receive lifestyle consultations or visit clinics. In our cohort, individuals with severe high blood pressure (SBP $\geq$ 160 mmHg or DBP $\geq$ 100 mmHg) or severe high blood glucose levels (FBG $\geq$ 130 mg/dL or HbA1c $\geq$ 7.0%) received a specially designed letter from their insurers urging them to visit a clinic. If a participant has an extremely severe glucose level (FBG $\geq$ 150 mg/dL or HbA1c $\geq$ 8.0%), insurers warn employers to encourage the relevant individual to visit a clinic. The algorithm of this national program by the Japanese Ministry of Health, Labour and Welfare and additional interventions for selected individuals are presented in **Fig 1**.

### Implementation of health guidance

Health guidance generally takes a form of a lifestyle consultation provided by healthcare professionals, including physicians, public health nurses, or nutritionists, unless their insurers instruct them to make direct clinic visits. Telephone-based follow-ups were conducted for participants who received lifestyle consultations. The Follow-up interval for each participant was determined according to the number of criteria for high CVD risk the participant met and the participant's age and smoking status. Insured members were able to select a hospital or clinic to receive health check-ups, lifestyle consultations, or direct clinic visits. Insurers are responsible for making health guidance invitations and providing their members with the opportunity to receive health guidance. Employers were recommended, not obliged by law, to support their employees in receiving health guidance, while providing their employees with the opportunity of attending check-ups was their obligation [17]. As Japan has an easy-access healthcare system [13], participants could usually visit doctors on weekdays (or Saturdays in some hospitals and clinics), even without appointments, in response to the health check-up results.

### Data source

We analyzed an anonymized insurer-based database by the Health Insurance Association for Architecture and Civil Engineering companies. This database includes information on employees in architecture or civil engineering companies and their family members who other insurers do not cover. We selected this insurer for the analysis because it is one of the largest employer-sponsored health insurers in Japan, covering over 400,000 individuals. In addition, the wording and design for the health guidance invitation remained consistent throughout the study period. The database contains demographic characteristics, medical history, and

## Health check-ups to identify individuals at high CVD risk

**Obesity criteria:**          **at least one**

- Waist ≥ 85 cm (men) or ≥ 90 cm (women)
- BMI ≥ 25 kg/m$^2$

**Other CVD risk factors:**          **at least one**

- High blood pressure:          SBP ≥ 130 mmHg or DBP ≥ 85 mmHg
- High blood glucose level:          FBG ≥ 100 mg/dL or  HbA1c ≥ 5.6%
- At risk of dyslipidemia:          TG ≥ 150 mg/dL or HDL ≤ 40 mg/dL

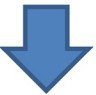

**Exclusion criteria:**
The use of medication for HTN, DM, or dyslipidemia.

## Health guidance:

- Lifestyle consultation by trained nurses
- Clinic visit

### Indications for additional interventions

- Specially designed letter urging participants to visit clinics:
    - Severe high blood pressure:          SBP ≥ 160 mmHg or DBP ≥ 100 mmHg
    - Severe high blood glucose level:     FBG ≥ 130 mg/dL or HbA1c ≥ 7.0%
- Warning for employers to encourage participants making clinic visits:
    - Extremely severe glucose level:     FBG ≥ 150 mg/dL or HbA1c ≥ 8.0%

**Fig 1. High-risk individual identification algorithm used in the annual health check-up program.** BMI = body-mass index, CVD = cardiovascular disease, DBP = diastolic blood pressure, DM = diabetes mellitus, FBG = fasting blood glucose, HbA1c = hemoglobin A1c, HDL = high-density lipoprotein cholesterol, HTN = hypertension, SBP = systolic blood pressure, TG = triglyceride.

laboratory testing data from health check-ups, self-reported lifestyle, lifestyle consultation records, and clinic visit records to treat HTN, DM, or dyslipidemia. We ascertained health guidance from a record of either lifestyle consultation or a clinic visit to treat HTN, DM, or dyslipidemia. Our preliminary analyses in the insurer's database confirmed that health check-up for each participant is conducted almost every 365 days, and health guidance is mostly conducted within 365 days from check-ups for each eligible participant (**see S1 and S2 Figs**). Data collection was conducted on February 14, 2021.

### Descriptive statistics

**Population, outcome, and statistical analysis.**   To investigate the adherence pattern to health check-ups and subsequent health guidance, we first described the following statistics among individuals eligible for health check-ups at the beginning of 2017 (n = 186,316):

1. The proportion of individuals who underwent health check-ups from April 1, 2017, to March 31, 2018 (i.e., the number of individuals who underwent health check-ups/ the number of individuals eligible for the health check-ups)

2. The proportion of individuals who were invited for health guidance (i.e., the number of individuals who were identified as having a high risk of CVD according to the health check-up and were not prescribed medications for HTN, DM, or dyslipidemia/ the number of individuals who underwent health check-ups in the fiscal year 2017)

3. The proportion of individuals who were followed up for 365 days after the check-ups (i.e., the number of individuals who underwent health check-ups in the fiscal year 2017, were identified as having a high risk of CVD, were not prescribed medications for HTN, DM, or dyslipidemia, and were followed up for 365 days/ the number of individuals who underwent health check-ups in the fiscal year 2017, were identified as high risk of CVD, and were not prescribed medications for HTN, DM, or dyslipidemia)

4. The proportion of individuals who received health guidance within 365 days of the health check-up (i.e., the number of individuals who received health guidance within 365 days due to high risk of CVD according to the health check-up/ the number of individuals who were invited for health guidance in the fiscal year 2017, were identified as having a high risk of CVD, were not prescribed medications for HTN, DM, or dyslipidemia, and were followed up for 365 days)

We defined the 4) above as an adherence to the health guidance invitation.

**Sensitivity analysis.** Initially, our primary analysis was performed using the dataset with individuals eligible for health check-ups at the beginning of 2017. We used single-year data for the analysis to avoid counting the same individual multiple times. The same analysis was repeated for different calendar years as a sensitivity analysis.

**Predictors of adherence to health guidance invitation.** *Population*. To identify predictors of improved adherence to health guidance invitation, we investigated 30,979 adults invited for health guidance in the fiscal year 2017. Since detailed data were only available among those who attended health check-ups, we explored predictors for adherence to health guidance invitations but not adherence to attending health check-ups. In addition, we excluded a relatively small proportion of individuals who were lost to follow-up for health guidance (785/ 30,979 [2.5%]). Thus, 30,194 adults were included in the main analysis.

## Potential predictors

We explored available variables as potential predictors: age (categorized into five-year age groups based on their age at the check-ups except that individuals who attended earlier than their 40th birthday in the fiscal year were also classified as 40–44); sex; insured status (main insured person or family member); self-reported behavioral patterns (smoking status [current smoking or not], alcohol consumption [no consumption, more than 60 g alcohol daily, or any other types of consumption], and exercise habits [more than 30 minutes of physical exercise on more than one day per week or not]); self-reported motivation for a lifestyle change (uninterested in, interested in, or starting lifestyle change); and blood pressure and laboratory results (categorized for HTN, DM, and dyslipidemia by the thresholds used to determine a specific health guidance invitation intensity in the program, as described in **Fig 1**). However, detailed information was unavailable on how these data were collected at each health check-up site, reflecting the nature of administrative data.

## Outcome

The outcome was defined as receiving health guidance, as previously described.

## Statistical analysis

The associations between potential predictors and adherence to health guidance invitations were evaluated using multivariable logistic regression analysis. As 16.7% (4,449/ 30,979) of the participants invited for health guidance had at least one missing variable, we implemented multiple imputations using the chained equation (MICE) method [18]. An interview with the database provider indicated that those missing values occurred independently from an individual's characteristics or actual values but based on which clinics or hospitals the check-ups took place. The imputation model included all variables in the logistic regression model and the information of the lost to follow-up. We generated five imputations with five imputation cycles.

In addition, we conducted two pre-specified sensitivity analyses. First, to account for attrition due to loss to follow-up, we performed the same regression with inverse probability of censoring weighting (IPCW) by calculating the probability of being lost to follow-up by logistic regression with the same predictor variables. The probability of censoring was calculated using a multivariable logistic regression model, which included all predictors noted under the "*Potential predictors*" sub-heading. Observations with missing values were excluded from the analysis. Second, we conducted the same regression with 'complete' cases with all necessary data and follow-up for 365 days. All analyses were performed using R 4.1.1.

## Ethics statements

The institutional review board of Kyoto University approved all study procedures and waived informed consent for participants (approval number: R0817) because all data used in the study were anonymized by a third-party data provider. The study did not access privately owned or protected land; nor sampled any protected species. All methods were conducted in accordance with the Helsinki Declaration.

## Results

### Descriptive statistics

A total of 186,316 individuals were eligible for health check-ups at the beginning of the fiscal year 2017. Fifty-eight percent of the participants were men, and most participants (89.5%) were aged under 65 years.

In 2017, 71.7% of the population (133,573/ 186,316) attended a health check-up (**Fig 2**). Among those 133,573 individuals, 25.7% (n = 34,294) were taking prescribed medication for HTN, DM, or dyslipidemia, and 23.2% (n = 30,979) were invited for health guidance as having a high CVD risk and without such medications; of these 30,979 invited for health guidance, 97.5% (n = 30,194) were followed up for 365 days from their check-ups. Of these 30,194 individuals, 35.2% (n = 10,614) received health guidance within 365 days.

The sensitivity analysis conducted for the years 2016 and 2018 showed similar descriptive statistics: 71.2% and 72.4% for the rate of attending health check-ups, 21.6% and 22.2% for the probability of being identified as at high risk of CVD and invited for health guidance, 97.2% and 97.6% for follow-up rates 365 days from check-ups, and 34.8% and 34.9% for the adherence rate of receiving health guidance, respectively.

### Predictors for adherence to health guidance invitation

**Primary analysis.** A total of 30,194 individuals invited for health guidance remained in the cohort for 365 days from their check-ups. Among them, 53% (16,046) were aged under 50 years, 90% (27,281) were men, and 62% (18,862) were non-smoker (**Table 1**). Individuals who

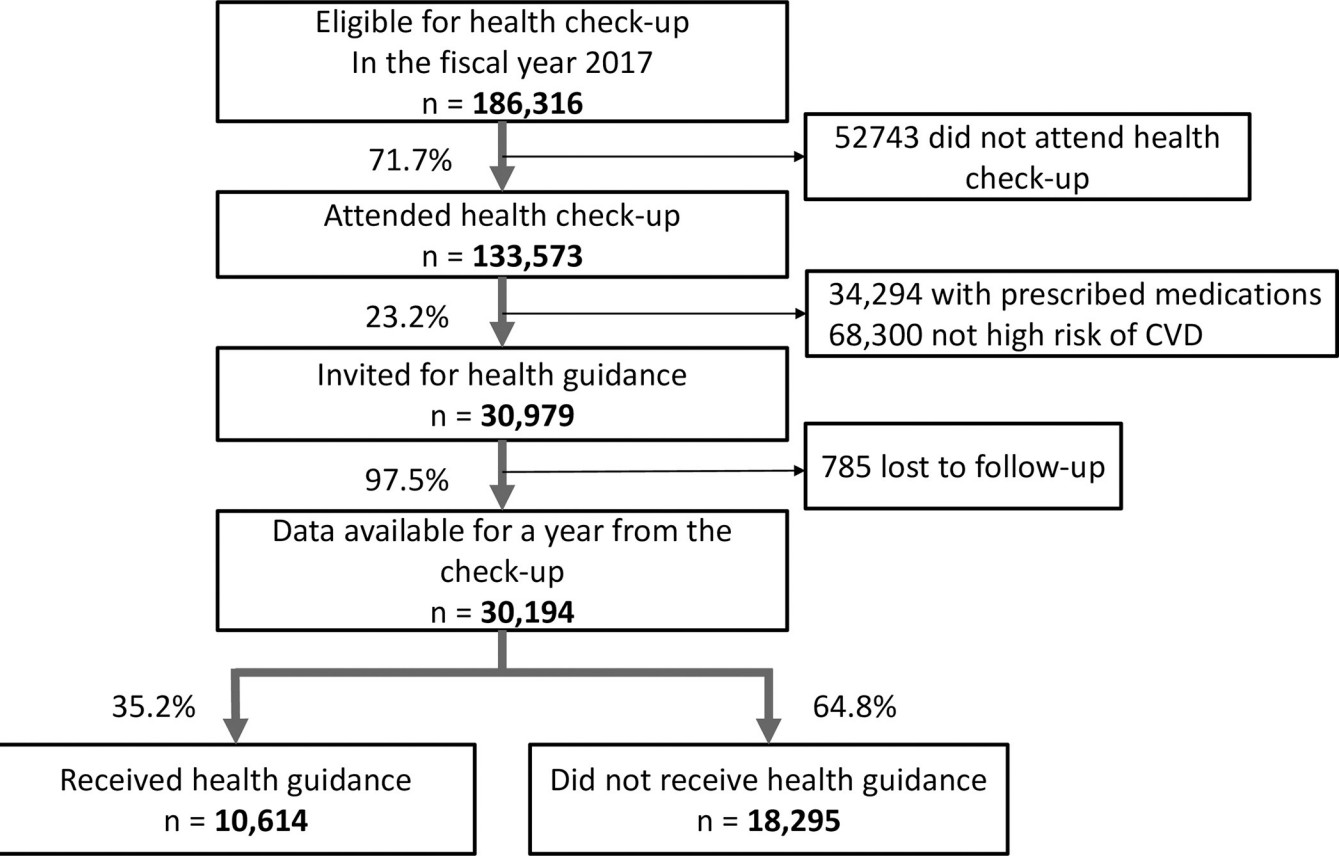

**Fig 2. Adherence flow in the national health check-up program and summary statistics of interest.** CVD = cardiovascular disease. Prescribed medication indicates medications to treat hypertension, diabetes, or dyslipidemia.

adhered to health guidance were more likely to be slightly older and have more concerning results in HTN and DM than individuals who did not adhere.

Age older than 50 years was significantly associated with a greater adherence probability, even after accounting for other potential predictors in a multivariable logistic regression model (e.g., odds ratio [OR] [95% confidence interval (CI)] for adherence in 65–69 and 70–74 vs 40–44 years old were 1.80 [1.59–2.03] and 2.06 [1.57–2.71], respectively) (**Fig 3**). More concerning results regarding the status of HTN or DM were also associated with higher odds of adherence. For example, individuals who received warnings from their employers because of their extremely high glucose levels showed an OR of 3.64 [3.08–4.30], while individuals who received a specially designed letter urging them to visit a clinic because of their high blood pressure and blood glucose level had ORs of 1.53 [1.34–1.73] and 1.90 [1.64–2.20], respectively. Alcohol consumption and smoking status were both associated with poor adherence. Those who were interested in lifestyle changes and those who started showed ORs of 1.24 [1.14–1.34] and 1.44 [1.28–1.63], respectively. In contrast, female and insured family members were associated with worse adherence to health guidance invitations.

**Sensitivity analysis.** Both pre-specified sensitivity analyses using the IPCW method and the complete case analysis provided similar results to the primary analysis (**Table 2**).

**Table 1. Cohort characteristics by adherence to health guidance invitation.**

| | Adhered | Not adhered | Overall |
|---|---|---|---|
| | (N = 10614) | (N = 19580) | (N = 30194) |
| **Age** | | | |
| 40–44 | 2281 (21%) | 4965 (25%) | 7246 (24%) |
| 45–49 | 2839 (27%) | 5961 (30%) | 8800 (29%) |
| 50–54 | 2175 (20%) | 4024 (21%) | 6199 (21%) |
| 55–59 | 1337 (13%) | 2124 (11%) | 3461 (11%) |
| 60–64 | 1303 (12%) | 1734 (9%) | 3037 (10%) |
| 65–69 | 583 (5%) | 673 (3%) | 1256 (4%) |
| 70–74 | 96 (1%) | 99 (1%) | 195 (1%) |
| **Gender** | | | |
| Male | 9661 (91%) | 17620 (90%) | 27281 (90%) |
| Female | 953 (9%) | 1960 (10%) | 2913 (10%) |
| **Insured status** | | | |
| Main insured | 10022 (94%) | 18310 (94%) | 28332 (94%) |
| Insured family member | 591 (6%) | 1263 (6%) | 1854 (6%) |
| Missing | 1 (0.0%) | 7 (0.0%) | 8 (0.0%) |
| **Recommendation intensity—HTN** | | | |
| None | 4002 (38%) | 8298 (42%) | 12300 (41%) |
| Mail post | 6142 (58%) | 10678 (55%) | 16820 (56%) |
| Specific letter | 470 (4%) | 604 (3%) | 1074 (4%) |
| **Recommendation intensity—DM** | | | |
| None | 4301 (41%) | 8479 (43%) | 12780 (42%) |
| Mail post | 5513 (52%) | 10497 (54%) | 16010 (53%) |
| Specific letter | 390 (4%) | 384 (2%) | 774 (3%) |
| Employer involvement | 410 (4%) | 220 (1%) | 630 (2%) |
| **Recommendation intensity—Dyslipidemia** | | | |
| None | 212 (2%) | 447 (2%) | 659 (2%) |
| Mail post | 10402 (98%) | 19133 (98%) | 29535 (98%) |
| **Smoking** | | | |
| Non-smoker | 7080 (67%) | 11782 (60%) | 18862 (62%) |
| Smoker | 3522 (33%) | 7782 (40%) | 11304 (37%) |
| Missing | 12 (0.1%) | 16 (0.1%) | 28 (0.1%) |
| **Alcohol consumption** | | | |
| Almost never | 2261 (21%) | 3955 (20%) | 6216 (21%) |
| Chance drink[a] | 5944 (56%) | 11078 (57%) | 17022 (56%) |
| Overdrink[b] | 1312 (12%) | 2484 (13%) | 3796 (13%) |
| Missing | 1097 (10.3%) | 2063 (10.5%) | 3160 (10.5%) |
| **Exercise habit** | | | |
| Insufficient | 7254 (68%) | 13340 (68%) | 20594 (68%) |
| >30min, >2days/week | 2043 (19%) | 3811 (19%) | 5854 (19%) |
| Missing | 1317 (12.4%) | 2429 (12.4%) | 3746 (12.4%) |
| **Self-reported motivation for a lifestyle change** | | | |
| No interest | 1443 (14%) | 3191 (16%) | 4634 (15%) |
| Interested | 5327 (50%) | 9895 (51%) | 15222 (50%) |
| Taking action | 2352 (22%) | 3723 (19%) | 6075 (20%) |
| Missing | 1492 (14.1%) | 2771 (14.2%) | 4263 (14.1%) |

HTN = hypertension, DM = diabetes mellitus.

a "Chance" alcohol consumption indicates any other style of alcohol consumption.

b "Overdrink" indicates everyday consumption of > 60g alcohol.

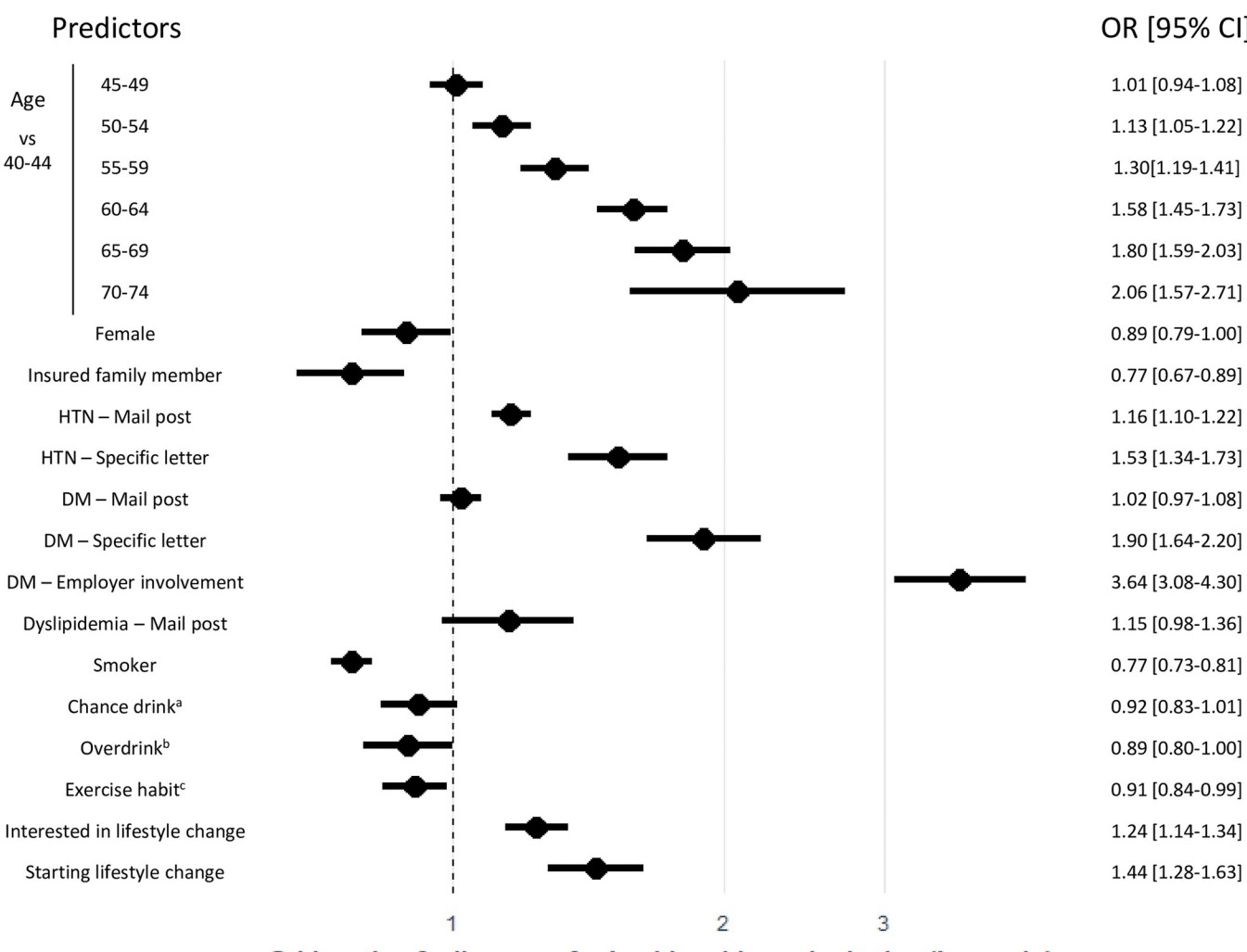

**Fig 3. Predictors for adherence to health guidance invitations.** CI = confidence interval, DM = diabetes mellitus, HTN = hypertension, OR = odds ratio. a "Overdrink" indicates everyday consumption of > 60g alcohol. b "Chance" alcohol consumption indicates any other style of alcohol consumption. c "Exercise habit" indicates more than 30 minutes of physical exercise in more than one day per week.

## Discussion

In the present study, we examined adherence patterns in Japan's nationwide health check-up program (*Tokutei Kenshin*) and investigated predictors of adherence among individuals at high risk of CVD. The adherence rate in eligible individuals was 71.7% for attending health check-ups but remained as low as 35.2% for health guidance invitations. Older age, more concerning results in terms of HTN, DM, and dyslipidemia, and self-reported motivation for lifestyle change were associated with improved adherence to health guidance invitation. These results were consistent in several sensitivity analyses.

We confirmed low adherence to health guidance invitations even though the program took place in a highly accessible healthcare system [13]. Given that the adherence rate to attending health check-ups was two times higher than to health guidance invitations, there should be some specific reasons for low adherence to health guidance invitations. The lack of employers' support would explain the low adherence because employers' obligation by law is to provide

**Table 2. Comparison of results between original and sensitivity analyses with different methods.**

| Predictors | Original[a] | Sensitivity Analyses | |
|---|---|---|---|
| | | IPCW[a] | Complete cases[a] |
| **Age (vs 40–44)** | | | |
| 45–49 | 1.01 [0.94–1.08] | 1.00 [0.93–1.07] | 1.00 [0.93–1.07] |
| 50–54 | 1.13 [1.05–1.22] | 1.14 [1.05–1.23] | 1.14 [1.06–1.24] |
| 55–59 | 1.30 [1.19–1.41] | 1.31 [1.20–1.44] | 1.30 [1.18–1.42] |
| 60–64 | 1.58 [1.45–1.73] | 1.64 [1.49–1.80] | 1.62 [1.47–1.78] |
| 65–69 | 1.80 [1.59–2.03] | 1.89 [1.65–2.17] | 1.83 [1.60–2.09] |
| 70–74 | 2.06 [1.57–2.71] | 2.25 [1.63–3.10] | 2.16 [1.60–2.90] |
| **Gender (vs Male)** | | | |
| Female | 0.89 [0.79–1.00] | 0.90 [0.79–1.01] | 0.89 [0.79–1.01] |
| **Insured status (vs Main insured person)** | | | |
| Insured family member | 0.77 [0.67–0.89] | 0.76 [0.65–0.88] | 0.77 [0.67–0.89] |
| **Recommendation intensity** | | | |
| HTN—Mail Post | 1.16 [1.10–1.22] | 1.17 [1.10–1.23] | 1.16 [1.10–1.23] |
| HTN—Urgent Letter | 1.53 [1.34–1.73] | 1.45 [1.26–1.68] | 1.46 [1.27–1.68] |
| DM—Mail Post | 1.02 [0.97–1.08] | 1.01 [0.96–1.07] | 1.01 [0.95–1.07] |
| DM—Urgent Letter | 1.90 [1.64–2.20] | 1.79 [1.52–2.11] | 1.84 [1.57–2.17] |
| DM—Employer Involvement | 3.64 [3.08–4.30] | 3.50 [2.89–4.24] | 3.49 [2.90–4.20] |
| DLP—Mail Post | 1.15 [0.98–1.36] | 1.17 [0.98–1.40] | 1.15 [0.96–1.37] |
| **Self-reported behavioral pattern** | | | |
| Smoker | 0.77 [0.73–0.81] | 0.77 [0.73–0.82] | 0.78 [0.74–0.82] |
| Chance drinking[b] | 0.92 [0.83–1.01] | 0.91 [0.85–0.97] | 0.91 [0.86–0.97] |
| Overdrinking[c] | 0.89 [0.80–1.00] | 0.90 [0.82–0.99] | 0.91 [0.83–0.99] |
| Exercise habit[d] | 0.91 [0.84–0.99] | 0.92 [0.86–0.98] | 0.92 [0.86–0.98] |
| Interested in lifestyle change | 1.24 [1.14–1.34] | 1.25 [1.16–1.34] | 1.25 [1.16–1.34] |
| Taking action on lifestyle change | 1.44 [1.28–1.63] | 1.42 [1.30–1.54] | 1.42 [1.31–1.54] |

DM = diabetes mellitus, HTN = hypertension, IPCW = Inverse probability-of-censoring weighting.

a Odds ratio for adherence [95% confidence interval].

b "Chance" alcohol drinking indicates any other style of alcohol consumption than overdrinking.

c "Overdrink" indicates everyday consumption of > 60g of alcohol.

d "Exercise habit" indicates more than 30 minutes of physical exercise in more than one day per week.

employees with opportunities for attending health check-ups but not for receiving health guidance [17]. In addition, adherence to health guidance invitations doubled when employers received warnings from insurers, compared to when only participants received specially designed letters urging direct clinic visits.

Older age was associated with higher adherence to health guidance invitations. More aged individuals would be more likely to pay attention to their health [19] and have higher chances of visiting clinics or hospitals for other reasons than younger adults [20]. Such visits may work as 'nudges' to receive health guidance. In addition, older adults may have fewer working hours than younger adults [21], resulting in more flexibility in their daily schedules to visit a healthcare facility.

The more concerning results regarding HTN, DM, or dyslipidemia were another predictor of improved adherence. This finding contrasts with previous research that identified good health status as a predictor of receiving health check-ups [3]. Both numeric information on those risk factors (e.g., blood pressure or laboratory test results) and the intensity of recommendation procedures (e.g., specific letter design and employer involvement) would have

contributed to better adherence in individuals with more concerning results. Indeed, Iizuka et al. [14] reported that information exceeding a specific clinical threshold independently improved adherence to health guidance invitations in this health check-up program.

Our findings indicate that future policy reforms designed to improve adherence to health check-up programs should target younger individuals, those with mild stages of HTN, DM, or dyslipidemia, and those with low motivation for a lifestyle change. Such interventions may include specific letter designs for selected individuals according to the predictors of adherence. Future studies on predictors from other aspects, such as socioeconomic or geographic status, are necessary to offer additional implications.

This study had some limitations. First, the generalizability of the study findings (particularly descriptive statistics) is limited, given that the study population consisted of workers in the field of architecture and civil engineering or their family members. The results may differ for workers in other fields with different sociodemographic backgrounds. Second, reflecting the nature of administrative data, there remains the possibility of misclassification of the study variables. For example, the observed clinic visits with relevant disease codes might not be in response to the health guidance invitation. Third, the definition of the high CVD risk in Japan's health check-up program, which emphasizes waist circumference, significantly differs from other commonly used risk prediction tools. For example, elevated low density lipoprotein cholesterol was not considered a high-risk indicator; and people with low or normal BMI with normal waist circumflex were excluded, though many of them may have had other CVD risk factors such as HT, DM, and dyslipidemia. This difference may limit the generalizability of our findings to individuals identified as high-risk using other risk prediction tools. Fourth, the study found a male predominance (90%) among those invited for health guidance, in contrast to the 58% male attendance at health check-ups. This deviation would also be attributed to the health check-up program's definition of high CVD risk. Last, our findings may be confounded by unmeasured or residual confounding factors.

## Conclusions

In Japan's nationwide health check-up program (*Tokutei Kenshin*), the adherence rate to attending check-ups was around 70%, but adherence to subsequent health guidance invitations remained at 35%. Older age, more concerning results in terms of HTN, DM, or dyslipidemia, and self-reported motivation for a lifestyle change were major predictors of improved adherence to health guidance invitation. Therefore, future policy reform to leverage adherence to this program should target younger individuals, those with mild stages of HTN, DM, or dyslipidemia, and those with low motivation for a lifestyle change.

## Supporting information

**S1 Fig. The interval between health check-ups in fiscal years 2017 and 2018 among adults who participated in both years.**
(DOCX)

**S2 Fig. Days between a health check-up in the fiscal year 2017 and health guidance for the first time since then among participants without prescribed medications for hypertension, diabetes mellitus, or dyslipidemia.**
(DOCX)

**S1 Table. The STROBE checklist.**
(DOCX)

## Acknowledgments

We would like to thank Editage [http://www.editage.com] for reviewing this manuscript in English.

## Author Contributions

**Conceptualization:** Yuichiro Mori, Shingo Fukuma.

**Data curation:** Yuichiro Mori.

**Formal analysis:** Yuichiro Mori.

**Funding acquisition:** Shingo Fukuma.

**Methodology:** Yuichiro Mori.

**Project administration:** Shingo Fukuma.

**Resources:** Shingo Fukuma.

**Supervision:** Kunihiro Matsushita, Kosuke Inoue, Shingo Fukuma.

**Writing – original draft:** Yuichiro Mori.

**Writing – review & editing:** Kunihiro Matsushita, Kosuke Inoue, Shingo Fukuma.

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
