## [Decision Letter · Decision Letter 0]

7 Feb 2023

PONE-D-22-26592Patterns and predictors of adherence to follow-up health guidance invitations in a general health check-up program in Japan: a nationwide cohort studyPLOS ONE

Dear Dr. Shingo Fukuma 

Thank you for submitting your manuscript to PLOS ONE. After careful consideration, we feel that it has merit but does not fully meet PLOS ONE’s publication criteria as it currently stands. Therefore, we invite you to submit a revised version of the manuscript that addresses the points raised during the review process.

We look forward to receiving your revised manuscript.

Kind regards,

Fred Nuwaha

Academic Editor

PLOS ONE

Journal Requirements:

Additional Editor Comments:

The Manuscript has been reviewed. There are major issues that have been identified and are attached.

Reviewers' comments:

Reviewer's Responses to Questions

**Comments to the Author**

1. Is the manuscript technically sound, and do the data support the conclusions?

Reviewer #1: Yes

2. Has the statistical analysis been performed appropriately and rigorously? 

Reviewer #1: Yes

3. Have the authors made all data underlying the findings in their manuscript fully available?

Reviewer #1: Yes

4. Is the manuscript presented in an intelligible fashion and written in standard English?

Reviewer #1: Yes

5. Review Comments to the Author

Reviewer #1: The authors describe adherence to a health screening programme among individuals at high risk of CVD in a nation-wide survey in Japan based on analysis of an insurance database.

In summary, 71.7% of eligible individuals attended a health check-up, 23.2% of them with high CVD risk and not on medication were invited for health guidance, and only 35.2% of them received health guidance. Older age, concerns regarding HTN, DM, and dyslipidemia, and self-reported motivation for lifestyle change were associated with improved adherence to health guidance invitation.

Comments:

1. Title: The study is not of the general population in Japan, but of a selected group of individuals (employees in architecture or civil engineering companies and their family members). The data source is an insurer database. The title needs to reflect these.

2. Why was an insurer database for employees in architecture or civil engineering companies and their family members selected? Why were other employment categories not included? This needs discussion/ justification.

3. High CVD risk was defined as obesity PLUS hypertension, diabetes/pre-diabetes, and high TG or low HDL. Elevated LDL was not considered as a high-risk indicator – why? People with low or normal BMI were excluded, and many of them may have had other CVD risk factors such as HT, DM, DL. This needs discussion/ justification.

4. Many other population-based studies and/or screening programmes use risk scores from a risk prediction tool (such as the Framingham score) to identify individuals at high CVD risk. How does the method used for detecting high risk individuals in the present study compare with these methods? Please discuss.

5. This data is unlikely to be representative of the Japanese population at large due to several reasons: 90% of study population were men; only one selected category of employees/ professionals (this is addressed by the authors); data source was the database of one main insurer, and there is no data on individuals with other insurers. How many of the Japanese population are registered with this insurer? These limitations need to be discussed.

6. Adherence to health guidance was ascertained from records of lifestyle consultation or clinic visit on the insurer database. There was no direct ascertainment from individuals. This is a limitation that can affect the accuracy of the findings. Needs discussion.

7. There is no discussion on how these findings compare with studies done elsewhere. The discussion needs to be expanded with references to such findings.

8. What are the strengths of the study, that would warrant its publication?

Minor comments:

Results: Pg 9: Line 245; says ‘….97.5% (n = 30,979) were followed up for 365 days …’. Is this correct? Or should it say (n = 30,194)? Please check.

6. PLOS authors have the option to publish the peer review history of their article (what does this mean?). If published, this will include your full peer review and any attached files.

Reviewer #1: No

---

## [Author Response · Author response to Decision Letter 0]

26 Apr 2023

Our responses are provided, along with comments from editors and reviewers.

1. Title: The study is not of the general population in Japan, but of a selected group of individuals (employees in architecture or civil engineering companies and their family members). The data source is an insurer database. The title needs to reflect these.

We appreciate the reasonable suggestion. We have modified the title accordingly. The revised title is "Patterns and predictors of adherence to follow-up health guidance invitations in a general health check-up program in Japan: a cohort study with an employer-sponsored insurer database".

2. Why was an insurer database for employees in architecture or civil engineering companies and their family members selected? Why were other employment categories not included? This needs discussion/ justification.

We appreciate the comment which clarifies the reason for the data source selection. We selected this insurer for the analysis because it is one of the largest employer-sponsored health insurers in Japan, covering 400 thousand of individuals. In addition, the wording and design for the health guidance invitation remained consistent throughout the study period. Therefore, we included this explanation in the Data source section in Methods.

"... This database includes information on employees in architecture or civil engineering companies and their family members who other insurers do not cover. We selected this insurer for the analysis because it is one of the largest employer-sponsored health insurers in Japan, covering 400 thousand of individuals. In addition, the wording and design for the health guidance invitation remained consistent throughout the study period. The database contains demographic characteristics, medical history, …"

3. High CVD risk was defined as obesity PLUS hypertension, diabetes/pre-diabetes, and high TG or low HDL. Elevated LDL was not considered as a high-risk indicator – why? People with low or normal BMI were excluded, and many of them may have had other CVD risk factors such as HT, DM, DL. This needs discussion/ justification.

We agree that the comment is a reasonable criticism of Japan's general health check-up program. However, the government has defined the definition of high CVD risk from the beginning of the program. Therefore, the definition was not under our control. Given the discrepancy between the cutoffs to predict high CVD risk between the program and commonly used prediction tools, we modified the limitation paragraph to reflect this issue. Please refer to the answer to comment 4.

4. Many other population-based studies and/or screening programmes use risk scores from a risk prediction tool (such as the Framingham score) to identify individuals at high CVD risk. How does the method used for detecting high risk individuals in the present study compare with these methods? Please discuss.

As per the answer to comment 3, the criteria to define high CVD risk significantly differ from other commonly used prediction tools. This difference may limit the generalizability of our findings to individuals identified as high-risk using other risk prediction tools. We edited the limitation paragraph to reflect this issue.

". Third, the definition of the high CVD risk in Japan's health check-up program, which emphasizes waist circumference, significantly differs from other commonly used risk prediction tools. This difference may limit the generalizability of our findings to individuals identified as high-risk using other risk prediction tools. Fourth, …"

5. This data is unlikely to be representative of the Japanese population at large due to several reasons: 90% of study population were men; only one selected category of employees/ professionals (this is addressed by the authors); data source was the database of one main insurer, and there is no data on individuals with other insurers. How many of the Japanese population are registered with this insurer? These limitations need to be discussed.

We appreciate the suggestion that the deviation of our cohort from the general population should be discussed. We want to note that the proportion of males in those who attended health check-ups was 58%, much closer to the general population than 90%. Therefore, we suggest that the main source of the deviation would be the definition of high CVD risk rather than population characteristics. We additionally discussed this issue in the limitation paragraph.

"… Fourth, the study found a male predominance (90%) among those invited for health guidance, in contrast to the 58% male attendance at health check-ups. This deviation would also be attributed to the health check-up program's definition of high CVD risk. …"

6. Adherence to health guidance was ascertained from records of lifestyle consultation or clinic visit on the insurer database. There was no direct ascertainment from individuals. This is a limitation that can affect the accuracy of the findings. Needs discussion.

We agree there would be misclassifications in the ascertained clinic visits in response to health guidance invitations. Therefore, we also have included this point in the limitation paragraph.

"Second, reflecting the nature of administrative data, there remains the possibility of misclassification of the study variables. For example, the observed clinic visits with relevant disease codes might not be in response to the health guidance invitation."

7. There is no discussion on how these findings compare with studies done elsewhere. The discussion needs to be expanded with references to such findings.

While there have been several studies on the predictors of receiving health check-ups, this study is the first, to our knowledge, to investigate the predictors of attending follow-up risk-stratified interventions after health check-ups. In light of this, we have revised the fourth paragraph of the discussion section to provide a comparison of our findings with previous research on predictors of receiving health check-ups. We acknowledge the valuable feedback received, which has enabled us to make our discussion more informative and precise.

"The more concerning results regarding HTN, DM, or dyslipidemia were another predictor of improved adherence. This finding contrasts with previous research that identified good health status as a predictor of receiving health check-ups (3). Both numeric information on those risk factors (e.g., blood pressure or laboratory test results) and the intensity of recommendation procedures (e.g., specific letter design and employer involvement) would have contributed to better adherence in individuals with more concerning results. …"

8. What are the strengths of the study, that would warrant its publication?

To our best knowledge, this study first examined adherence patterns and predictors of attending follow-up risk-stratified interventions after health check-ups. These findings have important implications for healthcare providers designing follow-up interventions after health check-ups, particularly for individuals with chronic conditions.

Minor comments:

Results: Pg 9: Line 245; says '….97.5% (n = 30,979) were followed up for 365 days …'. Is this correct? Or should it say (n = 30,194)? Please check.

Thank you for the correction. As the comment indicates, 97.5% (n = 30,194) were followed up for 365 days. We corrected the paragraph as suggested. The revised version of the paragraph is shown below. The number "35.2%" did not need correction as 10,614/30,194 equals to 35.15…, which can be rounded to 35.2.

"In 2017, 71.7% of the population (133,573/ 186,316) attended a health check-up (Figure 2). Among those 133,573 individuals, 25.7% (n = 34,294) were taking prescribed medication for HTN, DM, or dyslipidemia, and 23.2% (n = 30,979) were invited for health guidance as having a high CVD risk and without such medications; of these 30,979 invited for health guidance, 97.5% (n = 30,194) were followed up for 365 days from their check-ups. Of these 30,194 individuals, 35.2% (n = 10,614) received health guidance within 365 days."

---

## [Decision Letter · Decision Letter 1]

8 May 2023

PONE-D-22-26592R1Patterns and predictors of adherence to follow-up health guidance invitations in a general health check-up program in Japan: a cohort study with an employer-sponsored insurer databasePLOS ONE

Dear Dr. Fukuma, Thank you for submitting your manuscript to PLOS ONE. After careful consideration, we feel that it has merit but does not fully meet PLOS ONE’s publication criteria as it currently stands. Therefore, we invite you to submit a revised version of the manuscript that addresses the points raised during the review process.

We look forward to receiving your revised manuscript.

Kind regards,

Fred Nuwaha

Academic Editor

PLOS ONE

Journal Requirements:

Additional Editor Comments:

Please revise manuscript according to reviewers comments.

Reviewers' comments:

Reviewer's Responses to Questions

**Comments to the Author**

1. If the authors have adequately addressed your comments raised in a previous round of review and you feel that this manuscript is now acceptable for publication, you may indicate that here to bypass the “Comments to the Author” section, enter your conflict of interest statement in the “Confidential to Editor” section, and submit your "Accept" recommendation.

Reviewer #1: (No Response)

2. Is the manuscript technically sound, and do the data support the conclusions?

Reviewer #1: Yes

3. Has the statistical analysis been performed appropriately and rigorously? 

Reviewer #1: Yes

4. Have the authors made all data underlying the findings in their manuscript fully available?

Reviewer #1: Yes

5. Is the manuscript presented in an intelligible fashion and written in standard English?

Reviewer #1: Yes

6. Review Comments to the Author

Reviewer #1: The authors have addressed most of the comments.

They have partly discussed the limitations of the high CVD screening strategy used, but Comment 3 has not been specifically addressed in the discussion. Suggest add the following to the discussion as limitations of the screening strategy – (1) Elevated LDL was not considered a high-risk indicator; (2) People with low or normal BMI were excluded, and many of them may have had other CVD risk factors such as HT, DM, DL.

Minor comment:

Line 147 - suggest “ .. over 400,000…” instead of “… 400 thousand …”

7. PLOS authors have the option to publish the peer review history of their article (what does this mean?). If published, this will include your full peer review and any attached files.

Reviewer #1: No

---

## [Author Response · Author response to Decision Letter 1]

9 May 2023

1. They have partly discussed the limitations of the high CVD screening strategy used, but Comment 3 has not been specifically addressed in the discussion. Suggest add the following to the discussion as limitations of the screening strategy – (1) Elevated LDL was not considered a high-risk indicator; (2) People with low or normal BMI were excluded, and many of them may have had other CVD risk factors such as HT, DM, DL.

We appreciate the comment which clarifies the limitation of our manuscript. We modified the limitation paragraph reflecting the suggestion. 

"... Third, the definition of the high CVD risk in Japan’s health check-up program, which emphasizes waist circumference, significantly differs from other commonly used risk prediction tools. For example, elevated low density lipoprotein cholesterol was not considered a high-risk indicator; and people with low or normal BMI with normal waist circumflex were excluded, though many of them may have had other CVD risk factors such as HT, DM, and dyslipidemia. This difference may limit the generalizability of our findings to individuals identified as high-risk using other risk prediction tools. …"

2. Line 147 - suggest “ .. over 400,000…” instead of “… 400 thousand …”

We appreciate the suggestion that enhances the readability of our manuscript. We edited the line accordingly.

“… We selected this insurer for the analysis because it is one of the largest employer-sponsored health insurers in Japan, covering over 400,000 individuals. In addition, the wording and design for the health guidance invitation remained consistent throughout the study period. …” (Data source paragraph in Methods)

---

## [Decision Letter · Decision Letter 2]

15 May 2023

Patterns and predictors of adherence to follow-up health guidance invitations in a general health check-up program in Japan: a cohort study with an employer-sponsored insurer database

PONE-D-22-26592R2

Dear Dr. Shingo, 

We’re pleased to inform you that your manuscript has been judged scientifically suitable for publication and will be formally accepted for publication once it meets all outstanding technical requirements.

Kind regards,

Fred Nuwaha

Academic Editor

PLOS ONE

Additional Editor Comments (optional):

Reviewers' comments:

Reviewer's Responses to Questions

**Comments to the Author**

1. If the authors have adequately addressed your comments raised in a previous round of review and you feel that this manuscript is now acceptable for publication, you may indicate that here to bypass the “Comments to the Author” section, enter your conflict of interest statement in the “Confidential to Editor” section, and submit your "Accept" recommendation.

Reviewer #1: All comments have been addressed

2. Is the manuscript technically sound, and do the data support the conclusions?

Reviewer #1: Yes

3. Has the statistical analysis been performed appropriately and rigorously? 

Reviewer #1: I Don't Know

4. Have the authors made all data underlying the findings in their manuscript fully available?

Reviewer #1: No

5. Is the manuscript presented in an intelligible fashion and written in standard English?

Reviewer #1: Yes

6. Review Comments to the Author

Reviewer #1: The authors have satisfactorily addressed the review comments. I have no further revisions to suggest.

7. PLOS authors have the option to publish the peer review history of their article (what does this mean?). If published, this will include your full peer review and any attached files.

Reviewer #1: No

---

## [Editor Report · Acceptance letter]

17 May 2023

PONE-D-22-26592R2 

Patterns and predictors of adherence to follow-up health guidance invitations in a general health check-up program in Japan: A cohort study with an employer-sponsored insurer database 

Dear Dr. Fukuma:

I'm pleased to inform you that your manuscript has been deemed suitable for publication in PLOS ONE. Congratulations! Your manuscript is now with our production department. 

Kind regards, 

on behalf of

Dr. Fred Nuwaha 

Academic Editor

PLOS ONE